# Amyloidogenic Propensities of Ribosomal S1 Proteins: Bioinformatics Screening and Experimental Checking

**DOI:** 10.3390/ijms21155199

**Published:** 2020-07-22

**Authors:** Sergei Y. Grishin, Evgeniya I. Deryusheva, Andrey V. Machulin, Olga M. Selivanova, Anna V. Glyakina, Elena Y. Gorbunova, Leila G. Mustaeva, Vyacheslav N. Azev, Valentina V. Rekstina, Tatyana S. Kalebina, Alexey K. Surin, Oxana V. Galzitskaya

**Affiliations:** 1Institute of Protein Research, Russian Academy of Sciences, Pushchino 142290, Moscow Region, Russia; syugrishin@gmail.com (S.Y.G.); seliv@vega.protres.ru (O.M.S.); quark777a@gmail.com (A.V.G.); alan@vega.protres.ru (A.K.S.); 2Institute for Biological Instrumentation, Federal Research Center “Pushchino Scientific Center for Biological Research of the Russian Academy of Sciences, Pushchino 142290, Moscow Region, Russia; evgenia.deryusheva@gmail.com; 3Skryabin Institute of Biochemistry and Physiology of Microorganisms, Russian Academy of Sciences, Federal Research Center “Pushchino Scientific Center for Biological Research of the Russian Academy of Sciences, Pushchino 142290, Moscow Region, Russia; and.machul@gmail.com; 4Institute of Mathematical Problems of Biology, Russian Academy of Sciences, Keldysh Institute of Applied Mathematics, Russian Academy of Sciences, Pushchino 142290, Moscow Region, Russia; 5The Branch of the Institute of Bioorganic Chemistry, Russian Academy of Sciences, Pushchino 142290, Moscow Region, Russia; eyugorbunova@rambler.ru (E.Y.G.); mustaeva@rambler.ru (L.G.M.); viatcheslav.azev@bibch.ru (V.N.A.); 6Department of Molecular Biology, Faculty of Biology, Lomonosov Moscow State University, Moscow 119991, Russia; vrextina@gmail.com (V.V.R.); kalebina@gmail.com (T.S.K.); 7State Research Center for Applied Microbiology and Biotechnology, Obolensk 142279, Moscow Region, Russia; 8Institute of Theoretical and Experimental Biophysics, Russian Academy of Sciences, Pushchino 142290, Moscow Region, Russia

**Keywords:** fibrillogenesis, amyloidogenic regions, ribosomal S1 proteins, antimicrobial properties

## Abstract

Structural S1 domains belong to the superfamily of oligosaccharide/oligonucleotide-binding fold domains, which are highly conserved from prokaryotes to higher eukaryotes and able to function in RNA binding. An important feature of this family is the presence of several copies of the structural domain, the number of which is determined in a strictly limited range from one to six. Despite the strong tendency for the aggregation of several amyloidogenic regions in the family of the ribosomal S1 proteins, their fibril formation process is still poorly understood. Here, we combined computational and experimental approaches for studying some features of the amyloidogenic regions in this protein family. The FoldAmyloid, Waltz, PASTA 2.0 and Aggrescan programs were used to assess the amyloidogenic propensities in the ribosomal S1 proteins and to identify such regions in various structural domains. The thioflavin T fluorescence assay and electron microscopy were used to check the chosen amyloidogenic peptides’ ability to form fibrils. The bioinformatics tools were used to study the amyloidogenic propensities in 1331 ribosomal S1 proteins. We found that amyloidogenicity decreases with increasing sizes of proteins. Inside one domain, the amyloidogenicity is higher in the terminal parts. We selected and synthesized 11 amyloidogenic peptides from the *Escherichia coli* and *Thermus thermophilus* ribosomal S1 proteins and checked their ability to form amyloids using the thioflavin T fluorescence assay and electron microscopy. All 11 amyloidogenic peptides form amyloid-like fibrils. The described specific amyloidogenic regions are actually responsible for the fibrillogenesis process and may be potential targets for modulating the amyloid properties of bacterial ribosomal S1 proteins.

## 1. Introduction

The bacterial ribosomal S1 proteins are a unique protein family in which the number of structural S1 domain (one of the OB-fold, oligosaccharide/oligonucleotide-binding fold, domains) repeats varies in a strictly limited range from one to six. The multi-functional ribosomal S1 protein is a part of the 30S ribosomal subunit. It has a number of ribosomal functions, such as interaction with mRNA and tmRNA [1].

As we recently demonstrated, this family of polyfunctional ribosomal S1 proteins makes up about 20% of all bacterial proteins containing the S1 domain. Moreover, an automatic exhaustive analysis of ribosomal S1 sequences allowed us to demonstrate that the number of domains in S1 proteins is a distinctive characteristic for the phylogenetic grouping of bacteria in the main phyla [2]. 

The RNA-binding S1 domain is a β-barrel with an additional α-helix between the third and fourth β-strands [3]. A short helix in the S1 domain adjoins to the binding cleft in the protein structures and determines a strong preference for the single-stranded RNA (ssRNA). This helix distinguishes the S1 domains from other OB-fold proteins [4,5].

The S1 domain can be found in different eukaryotic protein families and protein complexes in different number variations [6]. An assessment of the susceptibility of the flexibility of S1 domains showed similar structural features and revealed functional flexible regions that are potentially involved in the interaction of natural binding partners. In addition, it was found that the central parts in multi-domain S1 proteins have the lowest percentage of flexibility. These facts allowed us to assume that the flexibility ratio in a separate domain is related to its role in S1 activity and functionality: a more stable and compact central part in multi-domain proteins is a vital for RNA interaction; terminal domains perform other functions [7,8]. Moreover, the obtained results correlate with the presence of S1 domain repeats in the S1 domain-containing proteins in the range from one (bacterial and archaeal) to fifteen (eukaryotic), and that is apparently due to the necessity of individual proteins to increase the affinity and specificity of binding to ligands [6].

It is known that amyloid-like fibrils of various proteins often have predominant β-strand elements in their structures [9]. For example, for the DNA-/RNA-binding protein YB-1 with disordered terminal domains, a tendency toward fibrillogenesis has been shown. Additionally, the obtained parameters of the protofibrils determine their packaging in the extended fibrils [10,11]. Note that the main structural element of the multifunctional eukaryotic protein YB-1 is the cold shock domain (CSD), which is the closest structural homologue of the S1 domain.

Our recent study of the full-sized ribosomal S1 protein, as well as its stable fragment (49 kDa) from *Thermus thermophilus*, showed that the disordered regions at the N- and C-ends of the protein can play a key role in its aggregation. Moreover, the stable S1 protein fragment (49 kDa) is less prone to aggregate formation than the whole protein from *T. thermophilus* [12,13].

Despite all these observations, a search for the amyloidogenic regions of S1 proteins and their propensity for fibrillogenesis has not yet been conducted. Here, we analyzed the tendency for the fibrillogenesis of bacterial ribosomal S1 proteins using a representative dataset, as well as the ability of individual amyloidogenic regions to form various fibril structures. The location and structure of the amyloid core may substantially depend on buffer conditions. According to the published data, bacterial cells growing at neutral pH maintain an intracellular pH of 7.2–7.8 [14,15]. It is known that many peptides and proteins are capable of forming amyloid-type aggregates at neutral pH values [16,17]. At the same time, the change in the pH of a solution can activate the fibrillation of proteins and affect the morphology of fibrils [18]. Ribosomal S1 protein acts as an RNA chaperone to unfold structured mRNA on the ribosome, which was studied at approximately neutral pH values of 7.5 [19]. Interestingly, changing the pH from neutral (pH 7) to acidic (pH 1–2) can switch “molecular chaperone” activity to “fibrillogenic” activity as shown for the heat shock protein HdeA from *E. coli* [20]. The change in pH from 7 to 2 did not activate fibrillation for the S1 protein (unpublished data), but as shown in the study, it affected the fibrillogenesis of peptides from the sequence of S1 from *E. coli* and *T. thermophilus*. Due to the fact that the pH value has a strong influence on the tendency for amyloid fibrillation, and the structure of formed amyloid fibrils depends on the pH, we focused on this dependence of the fibrillation processes [21]. We used the default software setting recommended by the authors of FoldAmyloid, PASTA 2.0, Waltz and Aggrescan [22,23,24,25] (a detailed description of the algorithms is given in the Materials and Methods section) for the theoretical identification of amyloidogenic regions in the separate structural S1 domains in the family of the ribosomal S1 proteins. 

## 2. Results

### 2.1. Analysis of Amyloidogenic Propensities of Ribosomal S1 Proteins

To analyze the amyloidogenic propensities of the full-sized bacterial S1 proteins, we used the FoldAmyloid, Waltz, PASTA 2.0 and Aggrescan programs, the possibilities and accuracy of which are described in [22,23,24,25]. The results are given in Table 1.

We evaluated the performances of the programs and revealed a relatively small average percentage of amyloidogenic regions. As seen, the percentage of amyloidogenic regions and the number of sequences varied between the different software packages. The difference in the algorithms underlying the operation of the used programs (Materials and Methods section, prediction and analysis of amyloidogenic regions) determines the difference in the percentages of the amyloidogenic regions for full-sized ribosomal S1 proteins. The number of sequences is varied due to the presence of sequences (in each studied group according to the domain number) in which the programs did not predict amyloidogenic regions. The highest percentage was found in one-domain S1 proteins (43%) using the Aggrescan program. The lowest percentage corresponded to the six-domain S1 proteins (4%) when the Waltz program was used. However, as noted above, the number of S1 repeats in the investigated proteins varies from one to six and strongly affects the protein size. The smallest size of the S1 domain was found in the members of the Mycoplasmatacea family (*Mycoplasma auris*—110 amino acid residues (a.a.) (UniProt ID: N9VCN6), *M. mobile*—116 a.a. (UniProt ID: Q6KH89)), and the largest size—876 a.a.—from *Salinibacter ruber* (UniProt ID: D5HA65) (Appendix A). 

The frequency distribution of the amyloidogenic regions in the ribosomal S1 proteins (according to the FoldAmyloid program) is shown in Figure 1a. Thus, the obtained results of the average percentage of amyloidogenic regions in the full-sized proteins containing different numbers of structural domains are ambiguous and may be misinterpreted (with the exception of the proteins containing one S1 domain).

In addition, it has been shown [12,13] that the recombinant ribosomal S1 protein from *T. thermophilus* and its stable fragment (49 kDa) tend to form only polymorphic aggregates. This feature may indicate the inability of amyloidogenic regions in a separate structural domain as part of a full-sized protein to form extended fibrils due to their strong involvement in the intermolecular domain interactions. Thus, the analysis of individual amyloidogenic regions in the separate structural S1 domains is effective for determining the features of the regions that may be involved in protein fibrillogenesis. The average percentage of amyloidogenic regions in each domain as well as for the full-sized proteins for individual structural bacterial S1 proteins is given in Figure 1b.

It was shown that some S1 domains have a higher average percentage of amyloidogenic regions than the full-sized proteins. Thus, to evaluate the number of amyloidogenic regions in a separate structural bacterial S1 domain, we also used the FoldAmyloid, Waltz, PASTA 2.0 and Aggrescan programs (Table 2).

We revealed that the percentages of amyloidogenic regions in the separate domains are closer in similarity (taking into account the deviations) than those for the full-sized proteins (Table 2). Thus, the highest percentage was found for one-domain containing proteins according to all the programs (30%—FoldAmyloid and Waltz, 46%—PASTA2.0 and 61%—Aggrescan). This fact is due to the presence of the ribosomal S1 protein sequence from *Mycoplasma mobile* (UniProt ID: Q6KH89) in this group, which has the highest percentage of the amyloidogenic regions among all the studied proteins (1331 records), more than 50% according to the used programs. The lowest percentages corresponded to the three-domain proteins (13%) when using the PASTA2.0 program (14% for the first domain in the three-domain proteins) and the Aggrescan program (24% for the second domain in the three-domain proteins). For these domains, the FoldAmyloid and Waltz programs also predicted about 15–20%. Moreover, the first S1 domain in six-domain proteins was identified as the domain having the smallest percentage of amyloidogenic regions (11–12%) according to the FoldAmyloid and Waltz programs. In general, it can be seen that the predictions of programs for different domains vary greatly and are difficult to interpret. Therefore, it is more appropriate to study the actual distribution of amylodoigenic regions in each structural domain.

### 2.2. Analysis of Distribution of Amyloidogenic Regions in Ribosomal S1 Proteins 

The real frequency distributions of amyloidogenic regions in the separate S1 domains (according to the FoldAmyloid, Waltz, PASTA 2.0 and Aggrescan programs) are shown in Figure 2.

As seen from Figure 2, the frequency of the distribution of amyloidogenic regions in the S1 domains allows the finding out of the characteristic features for each separate domain. The results of the used programs for the prediction of amyloidogenic S1 protein propensities vary greatly. Thus, the PASTA 2.0 and Aggrescan programs did not reveal clearly defined (with strict boundaries) amyloidogenic regions along the protein chain. At the same time, the FoldAmyloid and Waltz programs predicted certain regions with a large tendency for fibrillogenesis. Note that our recent study demonstrated that the sequence alignments of S1 proteins between separate domains in each group (different number of structural domains) revealed a rather low percentage of identity [2]. This fact explains the lack of strict boundaries of the amyloidogenic regions predicted by the Aggrescan program, which is based on the aggregation properties of individual a.a. residues. PASTA 2.0 controls the formation of β-sheets in proteins during the self-assembly of β-sheets in amyloid aggregates such as a cross-β structure. However, separate amyloidogenic regions and self-assemblies of β-sheets during amyloid aggregation may behave in different ways. The Waltz software is based on the experimental existing libraries of amyloids, so the obtained results can be considered as reliable. FoldAmyloid employs the expected number of contacts and the probability of hydrogen bond formation. As result, for our studied dataset, FoldAmyloid predicted regions with strict boundaries in each separate domain. The approximate locations of regions within structural domains are 10–15 a.a., 25–30 a.a., 55–60 a.a. and 65–70 a.a. The presence of strictly bounded amyloidogenic regions (according to the FoldAmyloid and Waltz programs) for each group of S1 domains (containing different numbers of structural domain, Appendix A) and between these groups allows us to consider these regions as unique and most relevant for the further experimental study of their tendency for fibrillogenesis (*Analysis of Fluorescence of Thioflavin T and Data from Electron Microscopy* section). 

### 2.3. Correlation between Amyloidogenic Regions and Secondary Structures in S1 Proteins

As mentioned above (Introduction section), amyloid-like fibrils of different proteins mostly have β-strands in their structure [9]. The S1 domain is one of the structural versions of the OB-fold, which is represented by a β-barrel with an additional α-helix between the third and fourth β-strands. The correlation between the amyloidogenic regions and secondary structures of S1 proteins for one-, two- and three-domain S1 proteins is shown in Figure 3.

The same data for four-, five- and six-domain proteins are given in Appendix A. Recently, we have shown that, despite the fact that individual S1 structural domains have a rather low percentage of identity within and between domains, they form different groups (depending on the domain number), with the preservation of secondary structure elements and some domain characteristics (flexibility and compactness) within the domain [6,7]. As seen from Figure 3 and Appendix A, strict intersection between the amyloidogenic regions and β-strands was revealed for all the studied sequences. Thus, the topology of the amyloidogenic regions in a separate S1 domain can also be considered as conserved and directly related to the positions of β-strands along the protein chain.

### 2.4. Sequence Logos and Uniqueness of Amyloidogenic Regions in S1 Domain

As mentioned above (Table 2), the separate structural S1 domains contain 12–30% of amyloidogenic regions (FoldAmyloid). In this case, the amyloidogenic regions generally correspond to β-strands (Appendix A). The search for specific sequence logos in the S1 domain allowed us to identify specific residues that are characteristic of amyloidogenic regions within the S1 structural domains (Table 3, Appendix A).

In Table 3, only amyloidogenic regions that are characteristic of all S1 domains and have the highest distribution density of amyloidogenic residues are given. As can be seen, some of the amino acids that fall into the amyloidogenic regions are quite conserved. For example, motifs of the region 25–30 a.a. (counting from the beginning of the domain) of the fourth and fifth domains of five-domain proteins are identical: (E/D)**G**L**(I/V)**H(I/V). The terminal amyloidogenic regions (regions 65–70 aa) of all the domains often end with the **LSRR** combination. The pattern **(F/Y)G(V/A)F(I/V)** is typical for the region of 10–15 a.a. of the third domain of the three-domain proteins; the third and fourth domains of the four-domain proteins; the third, fourth and fifth domains of the five-domain proteins; and the fifth domain of the six-domain proteins. The peptides D9G^T^, D10G^T^, V10I^T^, I10D^E^ and T10E^E^ selected for experimental studies (Table 4) also contain part of this conserved motif (D9G^T^: DFGVFVNLG; D10G^T^: DFGIFIGLDG; V10I^T^: VTDFGVFVEI; I10D^E^: IVRGVVVAID and T10E^E^: TDYGCFVEIE). The motif D**(I/L)SW** of the 25–30 a.a. region is conserved for the third domain of the four-domain proteins, and the third and fifth domains of the six-domain proteins. The V10NV^E^ peptide (Table 4) also contains part of this motif: VHLSDISWNV. For the third of the three-domain proteins; the first, third and fourth domains of the four-domain proteins; all the domains of the five-domain proteins; and the first, third, fourth and fifth domains of the six-domain proteins, the 55–60 a.a. region (counting from the beginning of the domain) has the same motif, (**V/I/L)**X(**A/Y**)**V(V/I)L**. The peptides E10D^T^, D10F^E^ and V10V^E^ and peptide E10D^E^ (Table 4) contain part of this motif (E10D^E^: EMEVVVLNID; D10F^E^: DEITVKVLKF; V10V^E^: VVNVGDVVEV; peptide E10D^E^: EIAAVVLQVD). A search for homologous peptides (10 a.a.) using the BLAST server [27] showed that they are unique and typical only for the ribosomal S1 protein family. Thus, it can be assumed that the results obtained for the selected peptides (*Amyloidogenic Propensities of Linkers between Structural S1 Domains* section) may be characteristic of other conserved peptides of S1 domains belonging to different taxonomic phyla.

### 2.5. Amyloidogenic Propensities of Linkers between Structural S1 Domains

As was discussed in [6,7,13], linkers between separate structural S1 domains in the multi-domain bacterial S1 proteins as well as protein terminal parts can play an important role in the S1 protein functioning and folding. We used the FoldAmyloid program to analyze the amyloidogenic propensities of bacterial S1 protein linker regions with different numbers of structural domains. The results (Appendix A) did not reveal any strong correlations. Thus, the highest percentage of the amyloidogenic regions in linkers was found in the two-domain containing proteins at the C-termini (about 80%). The lowest percentage was observed in the five-domain proteins between the fourth and fifth domains (7%). In general, in the three-, four-, five- and six-domain proteins, the percent of amyloidogenic regions was slightly higher in the linkers (between the structural domains) than in the terminal parts. Investigation of the distribution of the secondary structure (with the JPred program [26]) showed the predominance of α-helix elements in the linkers of all the bacterial S1 proteins. This fact agrees well with our previous results in [2,6]. By the searching of specific sequence logos, we identified characteristic residues of amyloidogenic regions in the linkers between structural S1 domains (Appendix A). The most conserved residue of all the linkers was tryptophan (W). Note that W belongs to the residues that increase the propensity to amyloid formation according to the FoldAmyloid program (http://bioinfo.protres.ru/fold-amyloid/) [22]. This fact partially explains the high percentage of some linkers in multi-domain S1 proteins. Among the conserved residues, there are arginine (R) and valine (V). Another part of the conserved residues consists of proline (P) and lysine (K), which are characteristic of α-helix elements (Appendix A).

### 2.6. Analysis of Fluorescence of Thioflavin T and Data from Electron Microscopy

Recently, we have studied the tendency for aggregation and the formation of amyloid-like fibrils of the S1 protein from *T. thermophilus*. It is a model for studying the structure of the bacterial ribosome and its individual proteins. We have shown that the protein and its stable S1 fragment (49 kDa) do not form amyloid but only amorphous aggregates at pH 7.5 and different ionic strengths [13]. At the same time, a short protein (one domain) can induce a characteristic increase in the fluorescence intensity of thioflavin T (for example, S1 from *M. mobile*; data not shown). Subsequently, the synthesis of only short peptides and the verification of their aggregation properties demonstrated their ability to form fibrils. That is, we can assume that with an increase in the protein size, the tendency for fibrillogenesis decreases. It was previously shown that S1 (60 kDa) *T. thermophilus* has a high tendency to form aggregates in solutions [13,28]. The ability of S1 to associate is determined by the structural features and functions of this protein, in particular, the formation of homodimers and binding to ribosomal proteins in the 30S ribosomal subunit [1]. The elucidation of the structural features of aggregates (amyloid or non-amyloid) is of great importance for controlling the aggregation process [29].

Based on the results of the computational prediction of amyloidogenic regions (potential amyloidogenic determinants (PADs)), eleven peptides were selected: four peptides from the ribosomal S1 protein sequence from *T. thermophilus* and seven peptides from the sequence of the *E. coli* ribosomal S1 protein [21]. The peptides VVEGTVVEVT (Peptide 1, V10T^T^), DFGVFVNLG (Peptide 2, D9G^T^), VTDFGVFVEI (Peptide 3, V10I^T^), EMEVVVLNID (Peptide 4, E10D^T^), IVRGVVVAID (Peptide 1, I10D^E^), DEITVKVLKF (Peptide 2, D10F^E^), TDYGCFVEIE (Peptide 3, T10E^E^), VVNVGDVVEV (Peptide 4, V10V^E^), DFGIFIGLDG (Peptide 5, D10G^E^), VHLSDISWNV (Peptide 6, V10NV^E^) and EIAAVVLQVD (Peptide 7, E10D^E^) were synthesized. The ability of the peptides to form fibrils was studied using two pH conditions (7.5 and 2.0). The PAD-containing peptides, with the exception of E10D^T^ and T10E^E^, had the ability to form amyloid-like fibrils at pH 7.5, while E10D^T^ and T10E^E^ formed amyloid-like fibrils at pH 2.0 (Table 4). 

We studied the heat-induced aggregation of the peptides from the ribosomal S1 proteins from *E. coli* and *T. thermophilus* using an amyloid-specific thioflavin T assay. The results confirmed that the peptides have the ability to form amyloid fibrils and amorphous aggregates. Long-term incubation at pH values of 7.5 and 2.0 led to the formation of thioflavin T positive, β-sheet rich aggregates with fibrillar, amyloid-like morphology visible by electron microscopy (Figure 4 and Figure 5, Appendix A).

## 3. Discussion

Bacterial ribosomal S1 proteins are a unique family of proteins with structural repeats, which in the classical sense differentiate them from proteins with tandem repeats. As we showed in our previous work [2,7], the separate S1 domains have a fairly low percentage of identity among themselves, but the full-sized proteins have a similar organization of multiple domains containing S1 proteins. In addition, the ribosomal S1 proteins appear to be closed in a “beads-on-a-string” organization, with each repeat being folded into a globular domain [6]. S1 proteins have recently been investigated for a tendency to aggregate, but the role of separate individual amyloidogenic peptides from different structural S1 domains in this process remained unclear [13]. Furthermore, the aggregation conditions for both the S1 domains themselves and most S1 amyloidogenic peptides have not yet been selected. Charged groups located in the core of the protein globule or on the surface affect the protein’s performance of its biological functions [30]. It is known from the literature that a change in the pH of the solution stimulates protein conformational changes, which leads to aggregation [31,32]. The amyloidogenic peptides that we predicted form amyloids, which is associated with the presence of amyloidogenic motifs [33] and their high percentage compared to in the whole protein S1. For a protein containing several immunoglobulin-like (Ig-like) domains, it was shown that high identity may be associated with a higher probability of the misfolding of this protein and a tendency to aggregation, including fibrils [34,35]. The same rule can be true for S1 proteins. From an evolutionary point of view, the amyloidogenicity of proteins falls under elimination selection; therefore, multi-domain proteins, which are relatively large compared to peptides, have a lower tendency to aggregate [36,37]. In this work, our theoretical and experimental results showed that some unique (specific only for bacterial S1 protein sequences) peptides from separate S1 domains tend to form fibrils. It should also be noted that the methods used in our work for the analysis of thioflavin T (ThT) and transmission electron microscopy (TEM) (traditionally used in such studies) [38] can sometimes lead to false conclusions [39]. Note that the whole protein and its amyloidogenic fragments behave differently, which is consistent with our results obtained earlier in the study of the Aβ peptide and its fragments [40]. 

## 4. Materials and Methods

### 4.1. Construction of Ribosomal S1 Protein Dataset

The representative dataset of the records was selected as described in [2]. The investigated dataset consists of 1331 records (Appendix A).

### 4.2. Prediction and Analysis of Amyloidogenic Regions

For the prediction of the amyloidogenic regions the FoldAmyloid (http://bioinfo.protres.ru/fold-amyloid/) [22], Waltz (http://waltz.switchlab.org) [23], PASTA 2.0 (http://protein.bio.unipd.it/pasta2/) [24] and Aggrescan (http://bioinf.uab.es/aggrescan/) [25] programs were used.

FoldAmyloid employs the expected packing density of a.a. residues and the probability of hydrogen bond formation as the basis for the prediction of amyloidogenic regions, including those enriched with hydrophobic residues and polar a.a. residues in protein sequences. We used a threshold value above 21.4 for five consecutive residues [22].

PASTA 2.0 is a piece of software based on the idea that the mechanisms and basic molecular interactions that control the formation of β-sheets in proteins are also preserved during the self-assembly of β-sheets in amyloid aggregates, for example, in a cross-β structure. Peptide sequences are identified as “hot spots” if their energy score is below the default threshold value of −5 [41]. 

The Waltz software was developed by expanding existing libraries of amyloidogenic peptides to distinguish true amyloids from amorphous aggregates. Amyloidogenic regions with threshold values above 0 are considered as hits that correspond to «Best Overall Performance» [42].

The AGGRESCAN software also used an approach based on their experimental data to develop an algorithm for identifying the aggregation properties of individual a.a. residues. Amyloidogenic regions with average values of amino-acid tendency to aggregate greater than −0.02 are considered as hot spots [25]. 

The uniqueness of the predicted amyloidogenic regions was checked with the BLAST server (https://blast.ncbi.nlm.nih.gov/Blast.cgi) [27].

### 4.3. Alignment and Prediction of Secondary and Tertiary Structures

The Multiple Sequence Alignment was implemented by the Clustal Omega service (https://www.ebi.ac.uk/Tools/msa/clustalo/) [43]. In our work, the standard parameters for this program were used. For the prediction of secondary structure for each sequence in our dataset, the Jpred4 program (http://www.compbio.dundee.ac.uk/jpred/) [26] was used. For the prediction of tertiary structure for the third S1 domain from *E. coli*, the Robetta server was used (http://robetta.bakerlab.org/) [44]. The generation of sequence logos was performed by the WebLogo 3 server (http://weblogo.threeplusone.com) [45].

### 4.4. Realization

Algorithms for the search, collection, alignment, representation and analysis of the data were realized using the freely available programming language Python 3 (https://www.python.org/) as implemented in PyCharm v.2017 community edition (https://www.jetbrains.com/pycharm/). The results of the data were visualized using the Matplotlib library [46].

### 4.5. Chemical Synthesis of Peptides

Four peptides from the sequence of the ribosomal S1 protein from *T. thermophilus* and seven peptides from the sequence of the ribosomal S1 protein from *E. coli* comprising the predicted amyloidogenic regions were prepared by solid phase synthesis (Table 4).

The chemical synthesis and purification of such peptides is a complex task due to the low solubility of the peptides in water and organic solvents. The synthesis was performed using the Fmoc methodology [48]. The purified peptide was tested using an Orbitrap Elite mass spectrometer (Thermo Scientific, Dreieich, Germany). The estimated peptide molecular weight coincided with the calculated one.

The chemical reagents used in the preparation of the amide forms of the peptides V10T^T^ and V10I^T^ were supplied by Fluka, IRIS Biotech GMBH and KHIMMED (Moscow, Russia). The reagents were used as received. Samples of the target peptides were prepared starting with aminomethyl polystyrene resin (1 g, 0.2 mmol/g initial loading) and Rink amide linker. The details of the solid phase peptide synthesis methodology employed in the preparation are described elsewhere [49,50,51], and these common protocols were followed throughout the synthesis. Whenever peptidyl polymer aggregation occurred, coupling protocols were modified either by employing various combinations of polar aprotic solvents [52] instead of DMF (dimethylformamide) or by using so called “chaotropic mixtures”, such as 4 M KSCN in DMF [53,54]. The crude peptide was released from the polymer with the concomitant removal of the acid-labile protecting groups using a common TFA(trifluoroacetic acid)-TIPS(triisopropylsilane)-H_2_O mixture at 95:2.5:2.5 (*v*/*v*/*v*) at retention time for 90 min. The crude peptides were purified by preparative HPLC, and the collected fractions were analyzed by mass spectrometry. Appropriate fractions were combined and lyophilized, providing the target peptide as a white, amorphous, non-hygroscopic solid. V10T^T^: yield 19 mg (ca. 11 %); for [M + H +] C_45_H_80_N_11_O_16_S, found 1030.5767, calculated 1030.5779; ∆M = 1 ppm. V10I^T^: yield, 18 mg (ca. 12 %); for C_54_H_82_N_11_O_15_S, found 1124.5988, calculated 1124.5986; ∆M < 1 ppm.

### 4.6. ThT Fluorescence Assay

The thioflavin T (ThT) assay was carried out essentially as described previously [55]. To facilitate dissolution, all the samples of peptides were initially dissolved in 100% DMSO, then the buffers 50 mM Tris-HCl, pH 7.5, 150 mM NaCl or 20% acetic acid pH 2.0, 150 mM NaCl were added (1% final concentration of DMSO) and filtered through 0.22 mm membranes to remove small aggregates. The kinetics of aggregation were studied by using the method of fluorescence spectroscopy. The fibril growth of peptides was monitored on the basis of ThT fluorescence. ThT is the most commonly used fluorescent dye for the following amyloid formation semi-quantitatively in vitro, specifically probing the fibrillar cross-β sheet content. Free ThT has excitation and emission maxima at ∼350 and ∼450 nm, respectively (upon binding to fibrils, the excitation and emission λmax change to ∼450 and ∼482–485 nm, respectively). A stock solution of ThT was prepared at a concentration of 20 mM in 50 mM Tris-HCl, pH 7.5, 150 mM NaCl or 20% acetic acid pH 2.0, 150 mM NaCl, respectively, and stored at 4 °C protected from light. The final concentration of ThT was 0.2 mM, and each peptide was 0.2 mM. ThT was present in the sample during the whole fibril formation reaction. Measurements of ThT fluorescence were performed on a Cary Eclipse fluorescence spectrophotometer (Varian, Mulgrave, Australia) in quartz cells of 3 × 3 mm at 37 °C. Each assay was carried out in triplicate.

### 4.7. Electron Microscopy

End-point solutions of peptides (0.2 mM in the buffer 50 mM Tris-HCl, pH 7.5, 150mM NaCl, 1% DMSO) incubated for 5 h with ThT were collected, and the samples were dissolved in the same solution to 0.2 mg/mL before studying with electron microscopy according to the previously described methods [55,56], with minor modifications. A formvar-coated copper grid 400 mesh (Electron Microscopy Sciences, Hatfield, PA, USA) was placed on a 10 μL sample. After 5 min of absorption, the grid with the preparation was negatively stained for 1.5–2.0 min with a 1% (weight/volume) aqueous solution of uranyl acetate. The excess of the staining agent was removed with filter paper. The preparations were analyzed using a JEM-100C (Jeol, Tokyo, Japan) transmission electron microscope at the accelerating voltage of 80 kV. Images were recorded on the Kodak electron image film (SO-163) at a nominal magnification of 40,000–60,000.

## 5. Conclusions

Using computational and experimental approaches, we analyzed the tendency for the fibrillogenesis of bacterial ribosomal S1 proteins, as well as the ability of individual amyloidogenic regions to form various fibril structures. Our study showed that some S1 domains have a higher average percentage of PAD than the full-sized proteins. Analysis of separate S1 domains (the amyloidogenic propensities, specificity of amino acid composition, and characteristics of secondary structures) allowed us to select 11 peptides for experimental study. We showed that the selected amyloidogenic regions may indeed be responsible for the fibrillogenesis process. The parameters of individual fibrils, apparently, are directly related to the amino acid sequence. For this protein family, such effects were shown for the first time. The described specific amyloidogenic regions may be potential targets for modulating the amyloid properties of bacterial ribosomal S1 proteins.

## Figures and Tables

**Figure 1 ijms-21-05199-f001:**
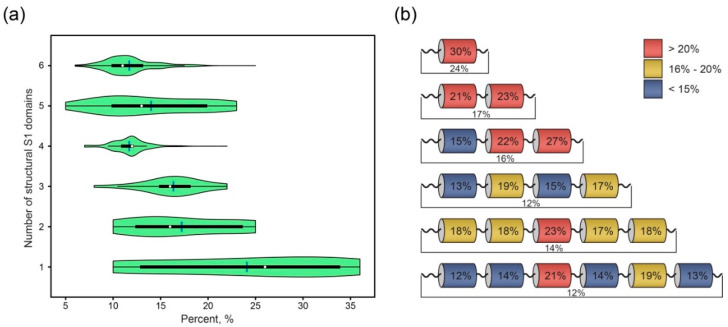
Amyloidogenic propensities of ribosomal proteins within the family of 30S ribosomal S1 proteins according to the FoldAmyloid program. (**a**) Violin plot of the frequency distribution of amyloidogenic regions in the ribosomal S1 proteins. Median: white dot; average: blue stripe; boxplot ends: first and third quantiles; thin black line: +/− the 1.5 fold interquartile range of values and outliers; probability density: green field. (**b**) Average percentage of amyloidogenic regions within each domain as well as for the full-sized S1 proteins.

**Figure 2 ijms-21-05199-f002:**
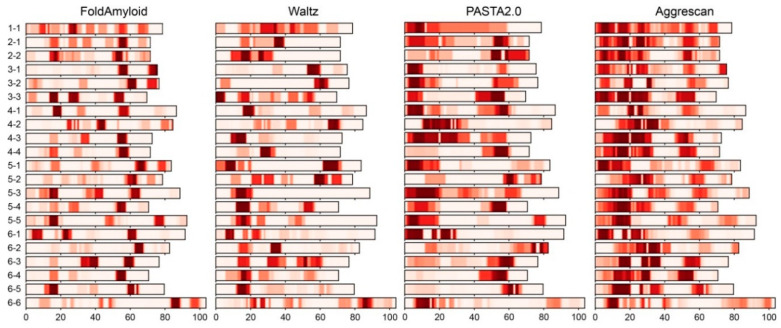
Distribution of amyloidogenic regions in the separate Appendix A domains according to the FoldAmyloid, Waltz, PASTA 2.0 and Aggrescan programs. The color scale corresponds to the frequency distribution of the amyloidogenic regions along the protein chain (aligned protein sequences). The size of each structural domain is about 70 amino acid residues (a.a.).

**Figure 3 ijms-21-05199-f003:**
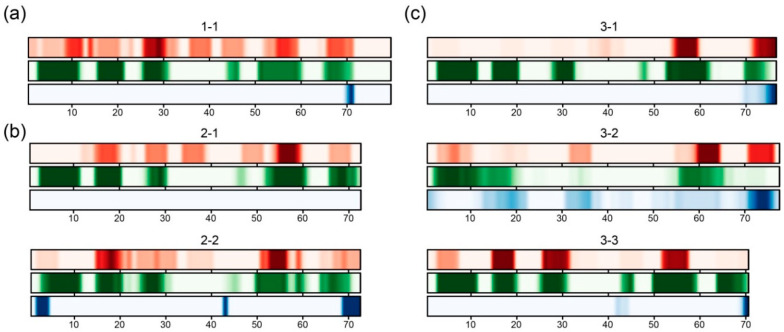
Distribution of amyloidogenic regions (orange) and secondary structures (green for β-strands; blue for α-helix) in the separate S1 domains according to FoldAmyloid for (**a**) one-domain S1 proteins, (**b**) two-domain S1 proteins (2-1 and 2-2) and (**c**) three-domain S1 proteins (3-1, 3-2 and 3-3). The color scale corresponds to the frequency distribution of residues along the protein chain (aligned protein sequences). The size of each structural domain is about 70 a.a. For the prediction of the secondary structure for each sequence in our dataset, the Jpred4 program [26] was used.

**Figure 4 ijms-21-05199-f004:**
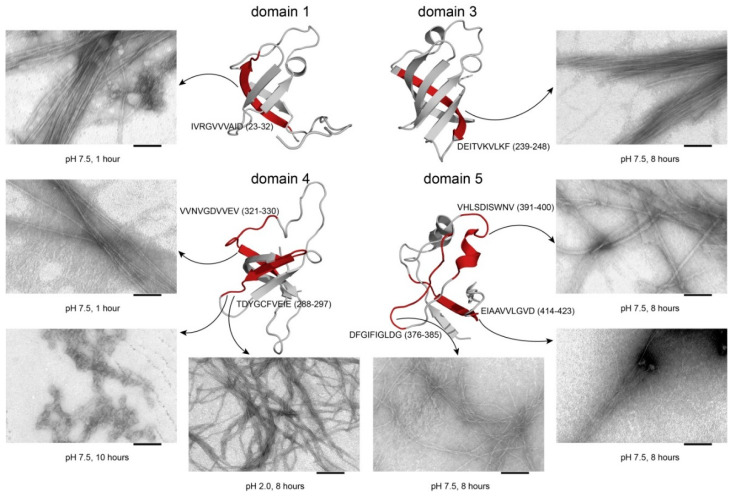
3D structures of the S1 domains from *E. coli*. Experimentally studied amyloidogenic regions (position and amino acid sequence are given) are highlighted with red color. Domain 1–PDB (Protein Data Bank, https://www.rcsb.org/) code: 2MFI; domain 4–PDB code: 2KHI; domain 5–PDB code: 5XQ5. 3D structure of domain 3 was predicted using the Robetta server. Scale bar = 100 nm.

**Figure 5 ijms-21-05199-f005:**
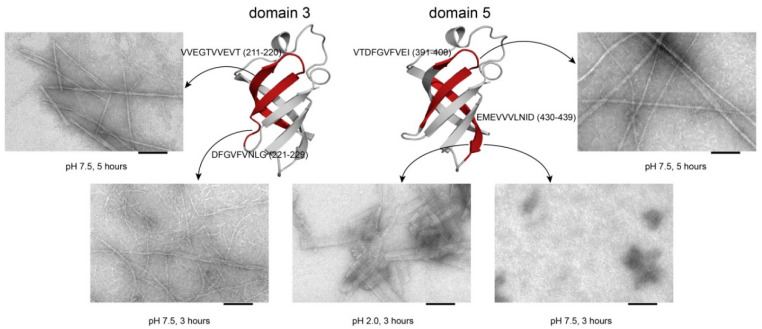
3D structures of the S1 domains from *T. thermophilus*. Experimentally studied amyloidogenic regions (position and sequence are given) are highlighted with red color. 3D structures of domains 3 and 5 were predicted using the Robetta server. Scale bar = 100 nm.

**Table 1 ijms-21-05199-t001:** Percentages of the amyloidogenic regions in the full-sized ribosomal S1 proteins. The largest and smallest values are highlighted in bold.

Amount of Structural S1 Domains	FoldAmyloid		Waltz		PASTA 2.0	AggreScan		
	% Amyloidogenic Regions	Number of Sequences	% Amyloidogenic Regions	Number of Sequences	% Amyloidogenic Regions	Number of Sequences	% Amyloidogenic Regions	Number of Sequences
1S1	24 ± 10	12	15 ± 13	7	29 ± 19	12	**43 ± 13**	12
2S1	17 ± 6	9	9 ± 6	6	16 ± 7	9	39 ± 10	9
3S1	16 ± 3	26	5 ± 3	26	10 ± 4	26	27 ± 4	26
4S1	12 ± 2	460	6 ± 2	458	13 ± 4	460	28 ± 6	459
5S1	14 ± 5	16	5 ± 3	15	10 ± 4	16	31 ± 9	15
6S1	12 ± 3	851	**4 ± 3**	816	8 ± 3	851	29 ± 5	848

**Table 2 ijms-21-05199-t002:** Percentages of the amyloidogenic regions in the separate S1 domains. The largest and smallest values are highlighted in bold.

Number of Structural S1 Domains	FoldAmyloid		Waltz		PASTA2.0		Aggrescan	
	% Amyloidogenic Regions	Number of Sequences	% Amyloidogenic Regions	Number of Sequences	% Amyloidogenic Regions	Number of Sequences	% Amyloidogenic Regions	Number of Sequences
1-1	**30 ± 15**	12	**30 ± 23**	5	**46 ± 30**	12	**61 ± 21**	12
2-1	21 ± 9	8	14 ± 10	5	28 ± 17	8	47±13	9
2-2	23 ± 8	9	15 ± 8	5	31 ± 13	5	54 ± 17	9
3-1	15 ± 4	25	13 ± 4	23	**14 ± 9**	20	35 ± 13	26
3-2	22 ± 8	26	15 ± 5	5	22 ± 10	8	**24 ± 10**	26
3-3	27 ± 6	26	13 ± 6	10	33 ± 10	22	56 ± 7	26
4-1	13 ± 6	453	12 ± 4	140	25 ± 12	252	28 ± 16	459
4-2	19 ± 6	459	13 ± 5	57	24 ± 14	294	36 ± 9	459
4-3	15 ± 4	456	14 ± 3	343	42 ± 20	355	49 ± 6	459
4-4	17 ± 5	460	14 ± 6	387	23 ± 9	331	46 ± 7	459
5-1	18 ± 7	15	16 ± 9	4	28 ± 11	11	38 ± 15	15
5-2	18 ± 9	15	12 ± 4	9	17 ± 8	4	38 ± 10	15
5-3	23 ± 11	15	14 ± 3	4	31 ± 16	13	43 ± 14	15
5-4	17 ± 7	16	13 ± 8	9	17 ± 5	7	42 ± 10	15
5-5	18 ± 8	16	11 ± 7	7	27 ± 12	8	38 ± 12	15
6-1	**12 ± 6**	637	**11 ± 5**	581	32 ± 8	715	35 ± 11	847
6-2	14 ± 8	847	14 ± 6	346	20 ± 11	159	38 ± 11	848
6-3	21 ± 5	851	13 ± 7	166	27 ± 16	329	43 ± 7	848
6-4	14 ± 6	850	13 ± 7	166	20 ± 7	457	43 ± 11	848
6-5	19 ± 6	848	13 ± 6	239	15 ± 8	266	39 ± 9	848
6-6	13 ± 6	524	13 ± 7	176	19 ± 10	310	25 ± 13	809

**Table 3 ijms-21-05199-t003:** Sequence logos of amyloidogenic regions in S1 domains.

Number of S1 Structural Domains	Position of the Amyloidogenic Regions in S1 Domains (a.a.)*			
	10–15	25–30	55–60	65–70
1-1	VXRY	(F/K)(G/C)(E/Y)(L/Y)	(V/E)KVL	L(L/V)(L/V)SFK
2-1	G(V/A/L)X(V/A)	(F/R)G(F/V)YP	EV(K/X)VL	R/G(L/V)(V/Y)LS
2-2	G(A/Y)EV(R/V)(L/Y)	(G/A)(L/F)(V/L)(H/P)	(V/E)(V/L)X(F/V)KV	(L/V)(S/H)X(K/R)
3-1			(E/H)(F/V)(F/L)(I/V)	(Q/M)(L/V)ILS
3-2		LRGFIP	(L/I)(P/T)(V/L)(K/A)FL	(K/R)LVLS
3-3	**(F/Y)G(V/A)F(I/V)**	GLLHIS	**(V/L)K(V/A)(M/L)(I/V)**	
4-1	(E/Q)(V/A)L(V/L)D	V(I/L)(P/T)(S/L)R	(V/I/L)(E/D)**(V/A/L)LV**	LSK(K/R)
4-2		L(I/V)(V/L)X(D/E)	(M/L)V(D/E)(M/T/H)R	(N/R)(V/E) (I/V)LSRR
4-3	**FGAF(I/V)**D	(E/Q)**(I/L)S(W**/H/Y)(K/E)	**(V/I)(**T/E/K)**V**(E/K) **(I/V)L**	
4-4	**FG(V/A)F(I/V)**	LVH(I/V)S	**(V/I/L)XV**K**(V/I)(I/L)**	
5-1	VXVDI	(G/A)X(V/I/L)PL	DX**(V/I/L)X(V/A/L)(**Y/Q)V	GX(I/Y)(L/Y)LSR
5-2	(I/V)X (G/V)X(I/V)	G(I/V)R(G/A)F(M/L)	**(V/L)**E(F/**V**)K**(I/V)(I/L)**	(N/R)(V/I)(V/I)(L/Y)LSR
5-3	**(F/Y)G(V/A)F(I/V)**	(E/R)(**I/L)S(W**/H/Y)(K/R)	VX**(V/A)XV(I/L)**	
5-4	**(F/Y)GAF(I/V)**E	(E/D)GL(I/V)H(I/V)	VE**(V/A)X (V/I)/L**	
5-5	E**GXF(I/V)**(E/A)	(E/D)GL(I/V)H(I/V)	EXX**V(I/L)**	
6-1	(G/A)XV(I/V)	(V/A)(L/Y)(V/I)(D/N)(A/Y)	DX**(V/I)XV(A/V)(L/I/V)**	
6-2	GX(I/V)	GETV(D/E)	EFK(V/I)(I/L)K	
6-3	YG(A/V)D	LH(I/V)T**D(M/I/L)(A/S)W**(K/R)	**(V/I)X(V/A)K(V/I) (L**/I) (K/R)	
6-4	YG(A/C)FV	EGL(V/Y)H(V/I)	(V/E)**V(E/D)(V/A)(M/V)(V/I)L**	
6-5	**FG(V/I/L)F(I/V)**	**D(I/L)SW**	**(V/I/L)X(A/Y)V(V/I)L**	
6-6	VXGX(I/V)		VEAK(V/I/L)	RX(V/I/L)XLS(V/I)(K/R)

* - approximate locations of region within structural domains; X—any amino acid residue; conserved amino acid residues are highlighted in bold.

**Table 4 ijms-21-05199-t004:** List of amyloidogenic peptides for chemical synthesis.

Sequences, Company	Localization in S1 Sequence, Domain (D), Species	Peptides Mw *	Peptides pI **
Peptide 1, VVEGTVVEVT (V10T)^T^, 1	(211–220 a.a), D3, *T. thermophilus*	1031.2	3.5
Peptide 2, DFGVFVNLG (D9G)^T^, 2	(221–229 a.a.), D3, *T. thermophilus*	967.1	3.8
Peptide 3, VTDFGVFVEI (V10I)^T^, 1	(391–400 a.a.), D5, *T. thermophilus*	1125.3	3.5
Peptide 4, EMEVVVLNID (E10D)^T^, 2	(430–439), D5, *T. thermophilus*	1160.3	3.4
Peptide 1, IVRGVVVAID (I10D)^E^, 3	(23–32 a.a.), D1, *E. coli*	1040.3	6.3
Peptide 2, DEITVKVLKF (D10F)^E^, 3	(239–248 a.a.), D3, *E. coli*	1191.4	6.3
Peptide 3, TDYGCFVEIE (T10E)^E^, 3	(288–297 a.a.), D4, *E. coli*	1175.3	3.4
Peptide 4, VVNVGDVVEV (V10V)^E^, 3	(321–330 a.a.), D4, *E. coli*	1028.2	3.5
Peptide 5, DFGIFIGLDG (D10G)^E^, 3	(376–385 a.a.), D5, *E. coli*	1053.2	3.5
Peptide 6, VHLSDISWNV (V10NV)^E^,3	(391–400 a.a.), D5, *E. coli*	1169.3	5.5
Peptide 7, EIAAVVLQVD (E10D)^E^, 3	(414–423 a.a.), D5, *E. coli*	1055.6	3.5

1—The Branch of the Institute of Bioorganic Chemistry, Russian Academy of Sciences, Pushchino, Russia. 2—ELABSCIENCE, China (intermediary—“Immunotex”, LLC NPO, Stavropol, Russia). 3—IQChemical, Saint-Petersburg, Russia. * and **—molecular weight and isoelectric point were calculated by the ExPASy server (https://web.expasy.org/cgi-bin/peptide_mass/peptide-mass.pl) [47]. ^T^— peptide from the sequence of the ribosomal S1 protein from *T. thermophilus*. ^E^— peptide from the sequence of the ribosomal S1 protein from *E. coli*.

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
