# Peer review of "Amyloidogenic Propensities of Ribosomal S1 Proteins: Bioinformatics Screening and Experimental Checking"

_ijms, 2020, doi:10.3390/ijms21155199_

Round 1
Reviewer 1 Report
This article combines both computational and experimental approaches to evaluate, analyze, and characterize the bacterial ribosomal S1 proteins and identify the key regions (11 peptides) which are responsible for the fibrillogenesis process. However, there are several major issues in this paper. First, the authors introduced several computational programs for their analysis, but they didn't explain what they are. It is important to clarify the algorithms and the tools that they used here, especially the results from these programs are very different. It is difficult for readers to understand the computational approaches. Second, several references are missing, and the authors need to carefully explain the data, figures, tables, and details. Last, the authors have to explain better what the functions of bacterial ribosomal S1 proteins are and why they are relating the amyloidogenic peptides to antimicrobial activities. There are several sentences that are confusing. Therefore, the current paper cannot be accepted.
Page 2 - Line 52. Please specify what the full name of "OB-fold" is. Is it "oligonucleotide/oligosaccharide-binding fold"?
Page 2 - Line 51-56. Please clarify what the roles of bacterial ribosomal S1 proteins are. What are their functions in bacteria?
Page 2 - Line 69-70 - sentence "in the range from one (bacterial and archaeal) to 15 (eukaryotic)...". I suggest to either use "from 1 (bacterial and archaeal) to 15 (eukaryotic)..." or "from one (bacterial and archaeal) to fifteen (eukaryotic)..."
Page 2 - Line 85-88. What is the native condition of the ribosomal S1 proteins in the bacteria? What is the pH value of bacteria? Could the authors please speculate it and clarify why the pH value is important here?
Page 2 - Line 88-91. Why did the authors choose these software programs? Could the authors please specify what they are? It is unclear to say they are recommended by the authors. In addition, please provide their references.
Page 3 - Table 1. The percentage of amyloidogenic regions and the number of sequences are varied between different software packages. I would suggest the authors summarize their algorithms and their differences. It will help the readers to better understand this table.
Page 4 - Line 125-128 and Table 2. What is the difference between Table 1 and Table 2? The sentence is unclear and needs more clarification.
Page 5 - Line 154-164 and Figure 2. The results of these programs are very different. Which one is more reliable? I highly recommend the authors to clarify this section 2.2 and discuss the figure. The figure is confusing.
Page 6 - Line 166-170 and Figure 3. Where did the secondary structures come from? How did the authors measure it? Are they measured from the experiments or predicted by the program? If the secondary structures were predicted by the program, please validate it with some experiments.
Page 7 - Line 209. Please provide the reference for the BLAST server.
Page 7 - Line 220. The authors first tried several programs, but they only used the FoldAmyloid program to analyze the amyloidogenic propensities here. Could the authors explain it? Is the FoldAmyloid program more accurate than others?
Page 8 - Line 227. What is the JPred program? The authors introduced a new program but failed to clarify and provide a reference.
Page 8 - Line 233. The authors provided a link <http://bioinfo.protres.ru/fold-amyloid/> without mentioning its name and developers.
Page 8 - Line 243-246. If these sentences are correct, the native ribosomal S1 proteins are difficult to aggregate and form fibrils. Again, what are the functions of bacterial ribosomal S1 proteins? Why amyloidogenic propensities of the ribosomal S1 proteins are important?
Page 9 - Line 272-273. Why is it unique?
Page 10 - Line 282-283. What pH will cause the protein conformational changes and aggregation? A strong acid or strong base? Why? Please clarify this sentence.
Page 10 - Line 283-285. This sentence is opposite to the previous sentence saying the change of pH (not sure which pH condition) can stimulate the conformational changes and lead to aggregation. Please clarify and explain.
Page 10 - Line 297-298. Why do the authors measure the antimicrobial activity of these peptides? These peptides were found from bacteria. Again, what are the roles of these proteins in nature? What is the relationship between antimicrobial activity and the ability to form fibrils?
Page 10 - Line 306-311. Please explain what these programs are and their algorithms. It is unclear here.
Author Response
Prof. Dr. Oxana Galzitskaya
Laboratory of Bioinformaics and Proteomics
Institute of Protein Research of the Russian Academy of Sciences
Institutskaya str.,4
Pushchino, Moscow region
142290 Russia
July 17, 2020
We are submitting here the revised manuscript of our paper entitled “Amyloidogenic Propensities of Ribosomal S1 Proteins: Bioinformatics Screening and Experimental Checking” (by Grishin Sergei Yu., Deryusheva Evgeniya I., Machulin Andrey V., Selivanova Olga M., Glyakina Anna V., Gorbunova Elena Yu., Mustaeva Leila G., Azev Vyacheslav N., Rekstina Valentina V., Kalebina Tatyana S., Surin Alexey K., Galzitskaya Oxana V). Please find attached the revised version of the manuscript and detailed response to the reviewers.
Reviewer: 1
This article combines both computational and experimental approaches to evaluate, analyze, and characterize the bacterial ribosomal S1 proteins and identify the key regions (11 peptides) which are responsible for the fibrillogenesis process. However, there are several major issues in this paper. First, the authors introduced several computational programs for their analysis, but they didn't explain what they are. It is important to clarify the algorithms and the tools that they used here, especially the results from these programs are very different. It is difficult for readers to understand the computational approaches. Second, several references are missing, and the authors need to carefully explain the data, figures, tables, and details. Last, the authors have to explain better what the functions of bacterial ribosomal S1 proteins are and why they are relating the amyloidogenic peptides to antimicrobial activities. There are several sentences that are confusing. Therefore, the current paper cannot be accepted.
Answer: We would like to thank reviewer 1 for work and valuable suggestions. We have changed the text of our manuscript according to the recommendations.
Page 2 - Line 52. Please specify what the full name of "OB-fold" is. Is it "oligonucleotide/oligosaccharide-binding fold"?
Answer: Abbreviation explanation has been added in the text, but there was explanation in the Abbreviation section (line 455).
Page 2 - Line 51-56. Please clarify what the roles of bacterial ribosomal S1 proteins are. What are their functions in bacteria?
Answer: Information about the roles of bacterial ribosomal S1 proteins and its functions in bacteria has been added in the text.
Page 2 - Line 69-70 - sentence "in the range from one (bacterial and archaeal) to 15 (eukaryotic)...". I suggest to either use "from 1 (bacterial and archaeal) to 15 (eukaryotic)..." or "from one (bacterial and archaeal) to fifteen (eukaryotic)..."
Answer: This sentence has been corrected.
Page 2 - Line 85-88. What is the native condition of the ribosomal S1 proteins in the bacteria? What is the pH value of bacteria? Could the authors please speculate it and clarify why the pH value is important here?
Answer: Thank you for your important question. According to the published data, bacterial cells growing at neutral pH maintain an intracellular pH of 7.2–7.8 (doi: 10.1128/AEM.00354-12, PMID: 6325389). It is known, that many peptides and proteins are capable of forming amyloid-type aggregates at neutral pH values (doi: 10.1074/jbc.M117.780528, doi: 10.1016/j.chembiol.2014.03.014.) At the same time, change in pH of solution can activate the fibrillation of proteins and affect the morphology of fibrils (doi: 10.3390/ijms18122551). Ribosomal S1 protein acts as an RNA chaperone to unfold structured mRNA on the ribosome, which was studied during neutral pH values 7.5 (doi: 10.1371/journal.pbio.1001731). Interestingly, changing pH from neutral (pH 7) to acidic (pH 1-2) can switch “molecular chaperone” activity to “fibrillogenic” activity as shown for heat shock protein HdeA E. coli (doi: 10.1074/jbc.RA118.005611). The change in pH from 7 to 2 did not activate fibrillation for S1 protein (unpublished data), but as shown in the study, it affected the fibrillogenesis of peptides from the sequence of S1 E. coli and T. thermophilus.
Page 2 - Line 88-91. Why did the authors choose these software programs? Could the authors please specify what they are? It is unclear to say they are recommended by the authors. In addition, please provide their references.
Answer: The corresponding references have been added. These programs have been chosen as the most frequently used for similar research in the analysis of amyloidogenic regions in proteins.
Page 3 - Table 1. The percentage of amyloidogenic regions and the number of sequences are varied between different software packages. I would suggest the authors summarize their algorithms and their differences. It will help the readers to better understand this table.
Answer: Really, the percentage of amyloidogenic regions and the number of sequences is varied between different software packages. The difference in the algorithms underlying the operation of the used programs (explanations for the algorithms have been added to the Materials and Methods section, Prediction and analysis of amyloidogenic regions) determines the difference in the percentages of amyloidogenic regions for full-sized ribosomal S1 proteins. The number of sequences is varied due to the presence of sequences (in each studied group according to the domain number) in which the used programs did not predict amyloidogenic regions. The corresponding explanation has been added in the text.
Page 4 - Line 125-128 and Table 2. What is the difference between Table 1 and Table 2? The sentence is unclear and needs more clarification.
Answer: The results from Table 1 relate to the percentages of the amyloidogenic regions in the full-sized ribosomal S1 proteins (for multi-domain containing proteins). The results from Table 2 relate to the percentages of the amyloidogenic regions in the separate S1 domains. The difference between results from Table 1 and Table 2 follows from their names and the text of the corresponding paragraphs.
Page 5 - Line 154-164 and Figure 2. The results of these programs are very different. Which one is more reliable? I highly recommend the authors to clarify this section 2.2 and discuss the figure. The figure is confusing.
Answer: Really, as seen from Figure 2, the results of used programs for the prediction of the amyloidogenic S1 protein propensities for each separate domain are very different. We have extended our discussion of Figure 2 in the text.
Page 6 - Line 166-170 and Figure 3. Where did the secondary structures come from? How did the authors measure it? Are they measured from the experiments or predicted by the program? If the secondary structures were predicted by the program, please validate it with some experiments.
Answer: The secondary structures for all studied ribosomal S1 sequences were predicted by the Jpred4 program (http://www.compbio.dundee.ac.uk/jpred/) [https://doi.org/10.1093/nar/gkv332]. The corresponding reference has been added in Figure 3.
Page 7 - Line 209. Please provide the reference for the BLAST server.
Answer: The corresponding reference has been added.
Page 7 - Line 220. The authors first tried several programs, but they only used the FoldAmyloid program to analyze the amyloidogenic propensities here. Could the authors explain it? Is the FoldAmyloid program more accurate than others?
Answer: One of the article authors (Prof. Dr. Oxana Galzitskaya) is one of developer the FoldAmyloid program [https://doi.org/10.1093/bioinformatics/btp691]. So, for analysis of the correlation between the amyloidogenic regions and secondary structures in S1 proteins (section 2.3), amyloidogenic propensities of linkers between structural S1 domains (section 2.5) and choice of the peptides for the experimental study it was decided to use our software (the FoldAmyloid program, which demonstrated good results for 14 years), to avoid multiple data and their interpretation when using several programs at the same time. The obtained fibrils as a result of experimental studies (section 2.6) indicate the high predictive ability of our software.
Page 8 - Line 227. What is the JPred program? The authors introduced a new program but failed to clarify and provide a reference.
Answer: The corresponding reference has been added in the text; description of the JPred program is given in the section Material and methods.
Page 8 - Line 233. The authors provided a link <http://bioinfo.protres.ru/fold-amyloid/> without mentioning its name and developers.
Answer: Link http://bioinfo.protres.ru/fold-amyloid/ relates to the FoldAmyloid program with the corresponding reference [Garbuzynskiy, S.O.; Lobanov, M.Y.; Galzitskaya, O. V. FoldAmyloid: a method of prediction of amyloidogenic regions from protein sequence. Bioinformatics 2010, 26, 326–332. https://doi.org/10.1093/bioinformatics/btp691]. The explanation was added in the text.
Page 8 - Line 243-246. If these sentences are correct, the native ribosomal S1 proteins are difficult to aggregate and form fibrils. Again, what are the functions of bacterial ribosomal S1 proteins? Why amyloidogenic propensities of the ribosomal S1 proteins are important?
Answer: It was previously shown that T. thermophilus S1 (60 kDa) has a high tendency to form aggregates in solutions (PMID: 17477047). The ability of S1 to associate is determined by the structural features and functions of this protein, in particular, formation of homodimers and binding to ribosomal proteins in the 30S ribosomal subunit (doi: 10.1093/emboj/19.23.6612). Elucidation of the structural features of aggregates (amyloid or non-amyloid) is of great importance for controlling the aggregation process (doi: 10.3390/ijms140612411).
Page 9 - Line 272-273. Why is it unique?
Answer: The chosen peptides from separate S1 domains may be considered as unique because they are specific only for bacterial S1 protein sequences. The corresponding explanation has been added in the text.
Page 10 - Line 282-283. What pH will cause the protein conformational changes and aggregation? A strong acid or strong base? Why? Please clarify this sentence.
Answer: The ability to fibrillation and structural differences in fibrils may be depending on the protonation state of N- and C-termini in the fibrils like insulin fibrils at strong acid or physiological pH conditions(https://www.mdpi.com/1422-0067/18/12/2551/htm). For a long time, we worked with insulin as a model protein for amyloidogenesis, and took for the present work the acid buffer conditions used for insulin fibrillation (https://doi.org/10.1080/19336896.2020.1776062). Acid pH is also necessary to neutralize the negative charge of Asp or Glu residues.
Page 10 - Line 283-285. This sentence is opposite to the previous sentence saying the change of pH (not sure which pH condition) can stimulate the conformational changes and lead to aggregation. Please clarify and explain.
Answer: Thanks. We have deleted this sentence.
Page 10 - Line 297-298. Why do the authors measure the antimicrobial activity of these peptides? These peptides were found from bacteria. Again, what are the roles of these proteins in nature? What is the relationship between antimicrobial activity and the ability to form fibrils?
Answer: We have deleted this speculation.
Page 10 - Line 306-311. Please explain what these programs are and their algorithms. It is unclear here.
Answer: Explanations for the algorithms of the used programs have been added to the Materials and Methods section, Prediction and analysis of amyloidogenic regions.

Reviewer 2 Report
Experiments are well designed and results clearly described. It is opinion of the reviewer that the manuscript can be accepted upon a minor revision. Some mistyping errors should be corrected through the whole manuscript. Moreover, authors should delete speculation on antimicrobial activity in "abstract" and "discussion" section, since they do not present results about this topic.
Author Response
Experiments are well designed and results clearly described. It is opinion of the reviewer that the manuscript can be accepted upon a minor revision. Some mistyping errors should be corrected through the whole manuscript. Moreover, authors should delete speculation on antimicrobial activity in "abstract" and "discussion" section, since they do not present results about this topic.
Answer: We would like to thank reviewer for valuable suggestions. We have changed the text of our manuscript according to the recommendations.
Round 2
Reviewer 1 Report
The authors have corrected my previous concerns. I would accept the current version.